# A Comparative Study of Lexical and Semantic Emoji Suggestion Systems

**ABSTRACT**

Emoji suggestion systems based on typed text have been proposed to encourage emoji usage and enrich text messaging; however, such systems' actual effects on the chat experience are unknown. We built an Android keyboard with both lexical (word-based) and semantic (meaning-based) emoji suggestion capabilities and compared these in two different studies. To investigate the effect of emoji suggestion in online conversations, we conducted a laboratory text-messaging study with 24 participants, and also a 15-day longitudinal field deployment with 18 participants. We found that lexical emoji suggestions increased emoji usage by 31.5% over a keyboard without suggestions, while semantic suggestions increased emoji usage by 125.1%. However, suggestion mechanisms did not affect the chatting experience significantly. From these studies, we formulate a set of design guidelines for future emoji suggestion systems that better support users' needs.

**Author Keywords**

Emoji suggestion; text entry; text messaging; mobile chat.

**ACM Classification Keywords**

Information systems~Texting; Human-centered computing~Text input

**INTRODUCTION**

Most forms of text-based computer-mediated communication (CMC) lack nonverbal expressions like vocal tones, facial expressions, and gestures that are useful in face-to-face conversations. However, several studies have shown that emojis can facilitate affective communication and serve a non-verbal function in text conversations [6,11,21]. Emojis are already widely used in text-based CMC, with nearly every instant messaging platform supporting their entry. Five billion emojis were sent per day on

| Sentence | Lexical | Semantic |
|---|---|---|
| I enjoyed the fish tonight very much! | 🐟🐠🦐🎏 | 😊😁😊😄❤️ |
| I love him but he just ignored me… | 😍😒🙂😔 😻😽❤️🍂 | ❤️😞😦😢😥 |
| I'm tired of "happy birthday" | 🍭🎁🎊🎉 | ✌️😫😑😐😥 |

**Table 1. Examples of lexical and semantic emoji prediction. With lexical prediction, the suggested emojis are related to the literal meaning of certain keywords. With semantic prediction, the suggestions focus on the meaning of the sentence.**

Facebook Messenger in 2017 [4], and half of all Instagram comments included an emoji as of mid-2015 [8].

Many mobile keyboards offer emojis as a set of pictographic Unicode characters. As there is a large and growing set of emojis, manually searching for and selecting emojis can be a tedious task interrupting the flow of text entry. Commercial products that automatically suggest emojis have helped the emoji entry process become more seamless [19,20]. These products usually come in two variations—lexical and semantic suggestions—as shown in Table 1. With lexical suggestions (*e.g.*, *Gboard*), relevant emojis appear in a candidate list based on recent keywords typed by the user. With semantic suggestions (*e.g.*, *Dango* [20]; Figure 1) proposed emojis are based on the meaning of the message's content rather than on specific keywords.

Although emojis themselves are known to enrich conversations [6,11], the role that different emoji suggestion systems play has not been explored. Instead, prior work on suggestion systems has focused on retrieval precision and recall [1,7,9]. But how do different suggestion mechanisms influence emoji usage? How do they differ in terms of usability? How do they influence the chat experience, specifically the *engagement* and the *clarity* of conversations?

To investigate these questions, we implemented a keyboard capable of offering both lexical and semantic emoji suggestions. We first evaluated the performance of the two suggestion mechanisms with an online study. The results

*Submitted for review.*

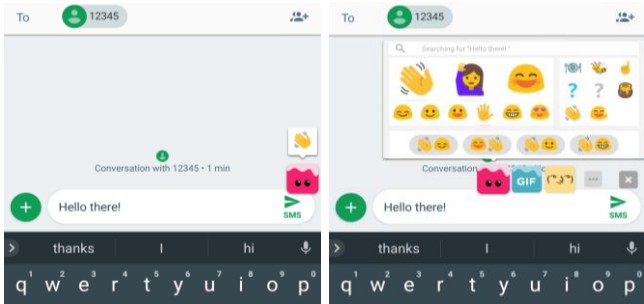

**Figure 1. The semantic emoji suggestion application Dango [20]. When text is typed, Dango pops up a suggested emoji based on semantic message content. The user can tap on an icon to see more options.**

confirmed that semantic suggestions were perceived as being more relevant than lexical suggestions. We then conducted an in-lab study with pairs of participants using three emoji suggestion mechanisms (no suggestions, lexical suggestions, and semantic suggestions). We also conducted a 15-day field deployment to evaluate emoji usage in real-world settings. We found that emoji suggestion systems increased emoji usage overall, with users picking more emojis via semantic suggestions versus lexical suggestions or no suggestions. Users also felt that semantic emoji suggestions were more relevant than lexical emoji suggestions. Although semantic suggestions received more positive feedback in terms of usability than lexical suggestions, neither had a significant effect on the participants' perceived chat experience overall.

The contributions of this work are: (1) results from an online study comparing the perceived relevance of lexical and semantic suggestions; (2) results from an in-lab study comparing emoji suggestion mechanisms within the mobile chat experience; (3) results from a longitudinal field deployment that tracked realistic usage of emoji suggestion systems; and (4) design guidelines of emoji suggestion systems based on the findings from our studies.

## RELATED WORK

Emoji-related research has become more prominent as emojis have grown in number and popularity. In this section, we review related work from three different areas: (1) emoji usage and its effects in online communication, (2) emoji entry techniques, and (3) the use of machine learning for producing semantic emoji suggestions.

### Emoji Usage in Online Communication

As Unicode character pictographs, emojis are treated similarly to other characters in text-based applications. In fact, emojis can even be used in text-only locales such as URLs. That being said, emojis represent richer information than plain text and are easier to share than images, giving emojis certain advantages over other forms of communication.

Emojis usage has steadily increased since emojis were introduced to the Unicode Standard in 2009. According to a report by Swiftkey in 2015 [5], their users inputted over one billion emojis in a four-month period. Although over 800 emojis were available to users during that time, traditional "face" emojis (*e.g.,* 😂) comprised nearly 60% of all emojis sent. Roughly 70% of the messages containing emojis expressed a positive emotion, and only 15% of the messages expressed a negative emotion.

Jain *et al*. [11] found that emojis are used to convey all kinds of emotions, and the number of emojis used in a message could determine the arousal of the sender. They also found that emoji combinations could be used to convey more complex expressions (*e.g.,* 😎⚽ meaning, "I'm relaxing and playing soccer"). Cramer *et al*. [6] conducted an online survey with 228 respondents, finding three major reasons for why people use emojis: (1) to provide additional emotional or situational information, (2) to change the tone of a message, and (3) to engage the recipient and maintain their relationship. Although every emoji has an intended definition, people also use emojis in highly personalized and contextualized ways to create "shared and secret uniqueness" [13,21]. Wiseman and Gould [21] cited an example where a couple used the pizza emoji (🍕) to express love because they both loved eating pizza.

There is no doubt that emojis extend and enrich the way people express themselves in text-based CMC. Our current work focuses on the role that suggestion mechanisms play in facilitating such expressions.

### Emoji Entry Techniques

Pohl *et al*. [14] provide a thorough review of emoji entry techniques. The most common entry method on current commercial keyboards is grouped enumeration, wherein users can scroll through different categories to select their emojis. However, as there are over 2,800 emojis,[1] so visually searching and selecting emojis is a tedious process. *EmojiZoom* [15] displays all emojis at once, requiring users to zoom to select one. However, this method still fails to scale as the number of emojis increases.

Querying techniques, such as text search or sketching, are implemented in many keyboards like *Gboard*. Users can search for emojis by sketching them or typing their intended meaning (*e.g.,* "happy birthday" for a cake emoji 🎂). Such techniques require users to have a target emoji in mind, and the process is slow.

Suggestion-based input methods have become popular in recent years. Lexical suggestions are offered by keyboards like the Apple iOS 11 keyboard. However, the suggestions do not work for all possible keywords, since keywords must be defined beforehand. For example, the pear emoji (🍐)

---

[1] https://emojipedia.org/stats/

appears in *Gboard*'s suggestion list when "pear" is typed, but it disappears if "pears" is typed.

A relatively new emoji suggestion technique that appears in products like *Dango* [20] uses semantic information. Instead of relying on keywords, semantic suggestion offers emojis based on the sentiment of the whole message. This mechanism often provides affective emojis like faces. Google deployed a similar system called *Smart Reply* [12]; rather than focusing on suggestions based on input, *SmartReply* auto-generates replies with emojis based on the context of the conversation.

### Producing Semantic Emoji Suggestions

To suggest emojis using semantics, emojis must be linked with the meanings of typed messages. Our keyboard implementation relies on a method from prior work called *DeepMoji* by Felbo *et al.* [9]. The implementation of their model used in this paper can be found on GitHub.[2]

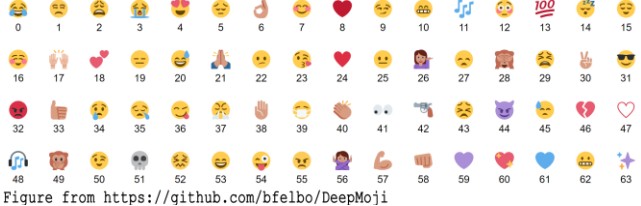

Figure from https://github.com/bfelbo/DeepMoji

**Figure 2. The 64 possible emoji suggestions of the semantic suggestion model used in our paper. Most of the emojis are faces, hearts, and hand gestures.**

DeepMoji uses a neural network to map textual features to relevant emojis. Felbo *et al.*'s dataset came from 1.2 billion tweets containing one of 64 common emojis (Figure 2). The reported top-5 suggestion accuracy of the model is 43.8%; in other words, roughly 2 of every 5 suggestions actually appeared in their test Twitter set. The model also reached 82.4% agreement on sentiment evaluation with humans on Amazon Mechanical Turk.

There were several limitations of Felbo *et al.*'s approach. First, the model was built for sentiment classification tasks, so most of the emojis it predicted were related to the messages' emotions rather than their meanings. If a user typed "happy birthday," for example, the predicted emojis would be happy face emojis (😊 🥴) rather than the birthday cake emoji (🎂). Second, the model was trained to handle only 64 emojis. Although those emojis were the most common on Twitter, many of them were faces rather than objects. Nevertheless, our goal is to examine the effects of different types of suggestion mechanisms rather than improving on the mechanisms themselves, so we did not extend the output emoji set beyond the original 64.

### EMOJI KEYBOARD DESIGN & IMPLEMENTATION

We built our Android keyboard using the open source project *AnySoftKeyboard*[3]. The keyboard interface is shown in

Figure 3. The keyboard uses the default auto-correction mechanism, but the word-suggestion feature is replaced with emoji suggestions. Users can enter special characters or numbers by tapping the upper-left button; they can enter emojis by tapping the lower-left button.

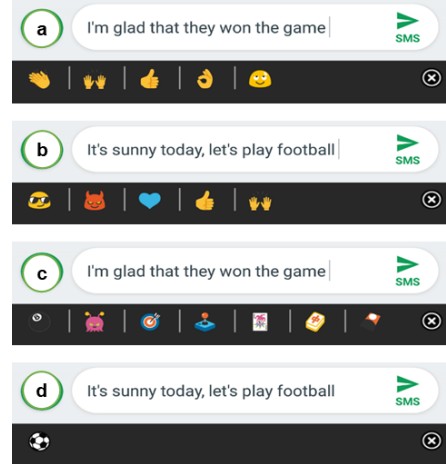

**Figure 3. (a, b) Semantic suggestion in our keyboard implementation always provides five emojis based on the message's content. (c, d) The number of emojis provided by lexical suggestion varies according to the number of keywords present. (c) If a keyword is related to many emojis, the user can scroll to select them. (d) When there is only one emoji related to the keyword "football," only that suggestion is shown.**

### Emoji Suggestion Mechanism

The overall text entry interaction of the keyboard is shown in Figure 4. As a user types in the text box, the keyboard provides word suggestions in the candidate list. When the user finishes typing a word, the keyboard suggests emojis instead of words in the candidate list. If the user picks an emoji from the list, it is inserted at the end of the message.

The suggestion result varies based on the mechanism in use. With semantic suggestion, the keyboard always presents five emojis after the user finishes typing a word. Suggestions are generated using the DeepMoji model [9] running on a remote server. The keyboard sends an HTTP POST request to the server each time the user finishes typing a word, and the server returns the top-five related emojis. (The amount of information transmitted is small and there were no latency concerns in our implementation or studies.)

With lexical suggestion, the keyboard suggests emojis only if the keyword list contains the last-typed word. If no emoji matches the last-typed word, the keyboard presents the most recent suggestions. For example, if the user types "football field," the keyboard will continue to suggest the football emoji (⚽) because there is no lexical match with the word "field." If no word anywhere in the message has a match in the emoji keyword list, the keyboard provides no suggestions. Lexical suggestion is implemented using the

---

[2]https://github.com/bfelbo/DeepMoji

[3] https://github.com/AnySoftKeyboard/AnySoftKeyboard

open-source emoji library *emojilib*[4]. The library provides a `*.json` file containing 1,502 emojis and their corresponding keywords. For example, the clapping emoji (👏) has words "hands," "praise," "applause," "congrats" and "yay."

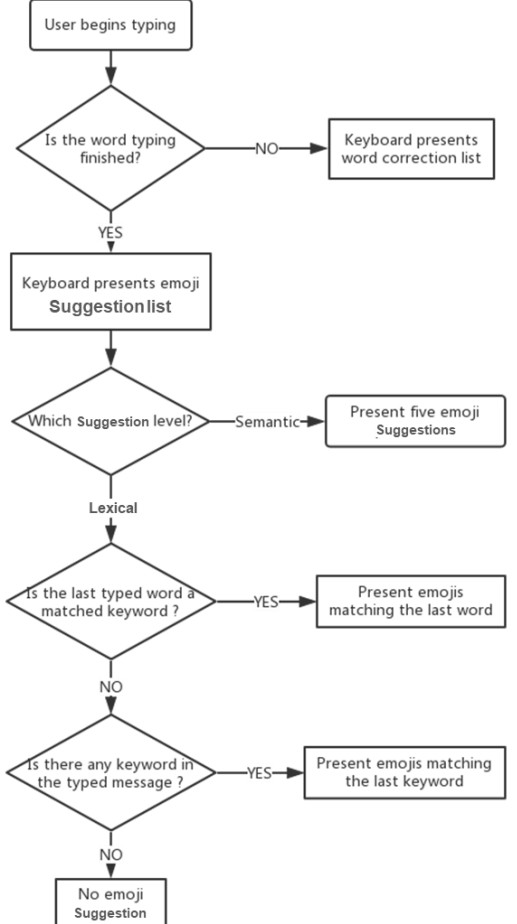

**Figure 4. A diagram of the text entry and emoji suggestion process with our keyboard.**

### Design Rationale

We did not force the frequency of emoji suggestion updates to be the same for both lexical and semantic suggestion mechanisms, as these mechanisms are fundamentally different in nature. For lexical suggestion, the opportune moment for updating the emoji suggestions is straightforward—whenever a keyword has been typed. For semantic suggestion, however, it is unclear when the suggestions should be updated because it is not obvious when the user is finished typing. Thus, our keyboard updates the emoji suggestions after the user finishes typing each word, not just keywords.

We also did not force an equal number of emoji suggestions across the two keyboards. Using a machine learning model for semantic suggestion returns a fixed number of emojis (five in *DeepMoji*), but lexical suggestion can produce a

variable number of emojis. Adding extra emojis when lexical suggestion produces too few emojis would confuse users with unrelated emojis, and conversely, trimming potentially relevant emojis from the semantic suggestions would make for a keyboard unrepresentative of its full potential.

### Data Logging

Our keyboard logs input statistics related to text and emoji entry: the number of typed characters, the number of deleted characters, the number of emojis manually added from the traditional emoji enumeration interface, and the number of emojis selected from the two suggestion lists. To respect participants' privacy, our keyboard did *not* log the content of any typed messages.

### PERCEIVED RELEVANCE OF EMOJIS FROM THE TWO SUGGESTION SCHEMES

Before having people use our keyboard, we conducted a preliminary crowd-sourced experiment to evaluate the performance of lexical and semantic emoji suggestion systems. Specifically, we were interested in quantifying the mechanisms' human-perceived accuracy.

We randomly sampled 50 tweets from the *Sentiment140* dataset[5], among which 25 contained positive sentiments and 25 contained negative sentiments. For each tweet, lexical and semantic emoji suggestions were generated and shuffled into a list. The tweets and their corresponding emoji suggestions were added to a single survey task on Amazon Mechanical Turk. Respondents were asked to "choose as many emoji options that made sense to be added at the end of each sentence." If the respondent felt that none of the emoji suggestions were relevant, they were allowed to select "none of the above." Respondents were not told that different suggestion mechanisms were in use.

We collected responses from 20 English-speaking Mechanical Turkers. The results are shown in Table 2. *Overall Relevance* refers to the number of selected emojis divided by the total number of emojis shown in the survey. There were 5,000 total emojis shown by the semantic suggestion system (50 tweets × 5 emojis per tweet × 20 people); respondents felt that 52.5% of them aligned with their corresponding tweet. For the lexical suggestion system, 2,520 emojis were shown, of which 21.6% were deemed relevant, less than half of those from the semantic suggestion system.

|  | Lexical suggestions | Semantic suggestions |
|---|---|---|
| Overall Relevance | 21.6% | 52.5% |
| Pick-1 Relevance | 32.6% | 94.6% |
| Top-1 Percentage | 6.0% | 94.0% |

**Table 2. Perceived relevance of the two suggestion systems, lexical and semantic. Higher percentages indicate greater perception of emoji relevance.**

---

[4] https://github.com/muan/emojilib

[5] http://help.sentiment140.com/home

*Pick-1 Relevance* examines whether any of the predicted emojis were selected for each tweet; in other words, as long as any emoji from a particular suggestion mechanism was deemed relevant for a tweet, the *Pick-1 Relevance* of that mechanism was 100% for that tweet. Out of 1,000 tweets (50 tweets × 20 users), semantic suggestion provided a relevant emoji 94.6% of the time. Lexical suggestion, on the other hand, only produced a relevant emoji 32.6% of the time.

*Top-1 Percentage* is a head-to-head comparison that captures whether the most commonly selected tweet per emoji came from the lexical or semantic suggestion system. Semantic suggestion was the overwhelming winner, providing the more relevant emoji for 47 of the 50 tweets.

The results show that Turkers generally perceived the emojis based on semantic content as more relevant than those based on keywords. This verified our hypothesis that the two suggestion mechanisms would be perceived differently. However, this study alone does not demonstrate how the different suggestion mechanisms affect the chatting experience, which leads us to our laboratory experiment.

## LABORATORY EXPERIMENT
Although in-lab experiments are not generally representative of realistic conditions, they are useful for studying conversations because they allow data to be gathered from both senders and receivers. An in-lab study enabled us to explore how the emoji suggestion mechanisms might affect the two conversational sides differently. If a person uses more emojis because they find it easier to do so, a recipient might react in two ways: they might enjoy the conversation more and reciprocate, or they might enjoy the conversation less due to annoyance, distraction, or confusion.

### Participants
Twenty-six participants (15 females, 11 males) between 18 and 34 years old (*M*=28.9, *SD*=4.2) were recruited via emails, word-of-mouth, and convenience sampling in a large university setting. The participants were randomly divided into 13 pairs. The pairs were constructed such that the participants did not know each other and did not meet face-to-face until the end of the study. Each participant was given $8 USD as compensation for the 30-minute study.

### Apparatus
Participants were provided with Nexus 6P smartphones running Google Android 7.0. Our keyboard was installed on each phone. *Wechat*[6] was used as the instant message application because *Wechat* provides a function to export the chat history. We used the chat history to verify the data logged by our mobile keyboard. Participants were instructed to avoid using *Wechat*'s built-in button for emoji entry since it bypassed our keyboard's logging functionality.

### Procedure
Participants were told that they would take part in an online chat experiment using our mobile keyboard. They chatted

with another participant for three 10-minute sessions, each of which was assigned to one of three emoji suggestion conditions: *no-suggestion*, *lexical*, or *semantic*. The order of the conditions was fully counterbalanced across participants. The participants were told that they could steer the conversation towards any topic of their choosing but were told that a "recent activity" could be used to start. The participants were also told that the only difference between the sessions would be the keyboard's emoji suggestion results, but they were not told anything about the suggestion mechanisms, what they were, or how they worked.

| Q1. Do you communicate online with your phone (SMS/IM/Email, etc.) a lot? |
|---|
| Q2. Do you use emojis in online conversations a lot? |

**Table 3. The questionnaire about online chatting and emoji use behavior.**

| Q1. The chatting experience was interesting. |
|---|
| Q2. My attention was focused on the conversation. |
| Q3. I could express my emotion clearly using the keyboard. |
| Q4. I felt constrained in the types of expressions I could |
| Q5. I was able to get an impression of my partner. |
| Q6. The chatting experience excites my curiosity. |

**Table 4. The survey questions about the chat experience. Answers were provided via Likert scales ranging from 1 (strongly disagree) to 7 (strongly agree).**

| Q1. I used the emoji suggestion a lot in my typing, and it was useful. |
|---|
| Q2. I would like to use this system frequently. |
| Q3. I thought the system was easy to use. |
| Q4. The system did well on proposing relevant emojis. |
| Q5. I like the emoji suggestion system better than the no-suggestion system. |

**Table 5. The usability survey for the suggestion keyboards. Answers were provided via Likert scales ranging from 1 (strongly disagree) to 7 (strongly agree).**

Before the conversation began, the participants were told to fill out a questionnaire that asked about their online chat and emoji use behaviors (Table 3). After each session, the participants filled out another questionnaire asking about their chat experience (Table 4). This questionnaire probed their engagement (Q1, Q2, and Q6) and perceived expressiveness and clarity (Q3, Q4, and Q5) regarding the chat experience. Both questionnaires were derived from prior work on CMC [17]. When participants used lexical or semantic suggestions in a session, they also completed the usability questionnaire shown in Table 5, which was adapted from the SUS survey [3]. At the end of the 30-minute session, participants were interviewed with two open-ended questions: (1) "How do you like the suggestion keyboards? Do you find they affect you (in negative or positive ways) in online chatting?" and (2) "Do you find any problems with

---

[6] https://www.wechat.com/en/

the keyboard suggestion mechanism, or do you have any suggestions?"

**Design & Analysis**
The study was a single-factor three-level within-subjects design with the suggestion mechanism as the independent variable: *no-suggestion*, *lexical*, and *semantic*. We utilized multiple statistical analyses according to the nature of the dependent variables: character count measures were analyzed using the aligned rank transform procedure [10,22]; emoji count measures fit a Poisson distribution, and were therefore analyzed with mixed model Poisson regression; Likert-scale responses were treated as ordinal measures and were therefore analyzed with mixed model ordinal logistic regression. Further specifics are given with each analysis in the results.

**RESULTS OF THE LABORATORY STUDY**
In this section, we describe the results of the study comparing the three levels of the *Suggestion* factor: no suggestions, lexical suggestions, and semantic suggestions.

During the study, one pair of participants did not conduct what we considered a realistic conversation. In one of their sessions, they sent only nonsensical numbers and capital letters to each other. This participant pair was therefore excluded from our analyses, and another pair was recruited in their place. Thus, our dataset included 12 valid participant pairs with two pairs per *Suggestion* order due to full counterbalancing (3! conditions = 6 orders). We collected 12×3=36 data logs of valid sessions, together with 72 surveys regarding the chat experience and 48 usability surveys for emoji suggestion. We conducted formal analysis with open coding, in which research team members identified any themes or codes they discovered from the 48 responses to the open-ended questions (Q1-Q2 in Table 3).

**Participant Phone Use**
Among the 24 participants, 22 stated that they *always* communicate with their phone, while the other two stated that they only used their phone *sometimes*. Nine participants stated that they *always* use emojis in online conversations, 14 *sometimes*, and one *seldom*. As for how our participants normally enter emojis, 14 participants manually selected emojis from a list, one participant used lexical suggestions from the keyboard, and nine used both methods.

**Count Measures**
The descriptive results of the logged data are shown in Table 6. *Total Characters* is the number of characters excluding emojis sent in the conversation; *Total Emojis* is the number of emojis used in the conversation, however they might have been inputted; and *Selected Emojis* is the number of emojis picked from the suggestion list.

|  | Total Characters | Total Emojis | Selected Emojis |
|---|---|---|---|
| No suggestions | 545.33 (211.58) | 2.17 (2.85) | N/A |
| Lexical | 542.04 (224.42) | 3.29 (3.51) | 0.88 (1.33) |
| Semantic | 579.79 (239.38) | 3.29 (2.93) | 2.17 (2.37) |

**Table 6. Means (and standard deviations) of *total characters, total emojis,* and *selected emojis* in three conditions.**

A non-parametric aligned rank transform [10,22] with a mixed model analysis of variance was performed on *Total Characters*. *Suggestion* had no significant effect on *Total Characters* ($F_{(2, 46)}$= 0.78, *n.s.*), indicating that the suggestion mechanism did not affect the overall volume of characters participants exchanged.

*Total Emojis* and *Selected Emojis* were conditionally fit to a Poisson distribution, as is common for count data [18], and mixed model Poisson regression was conducted on both measures. *Suggestion* had only a marginal effect on *Total Emojis* ($\chi^2_{(2,N=48)}$=5.25, *p*=.072). However, *Suggestion* did have a significant effect on *Selected Emojis* ($\chi^2_{(1,N=48)}$=7.76, *p*<.05), with semantic suggestion resulting in more selected emojis than lexical suggestion.[7] This result indicates that although the total number of emojis participants used across conditions was similar, participants selected more semantic-generated emojis than lexical-generated ones.

**Questionnaire Results**
Participants responded to the questionnaires along a 7-point Likert Scale (1=*strongly disagree,* 7=*strongly agree*), so the data were analyzed using mixed model ordinal logistic regression. Surprisingly, there were no significant results across the different *Suggestion* levels for any question regarding either the chat experience (Table 4) or usability (Table 5).

**Discussion of the Laboratory Study**
Based on the analysis of emoji counts in the study, we found that although different suggestion levels resulted in similar amounts of inputted emoji, participants tended to pick more from semantic suggestions than from lexical suggestions. One surprising finding was that although the usage of emojis indeed affected the senders' chat experience, the suggestion type did not affect the chat experience significantly. One explanation is that different suggestion mechanisms only affect *how* the user inputs emojis, rather than *what* they input. As long as they can input the expected emojis, the chat experience is not affected.

Looking at participants' interview answers, we found that participants did notice the difference between the suggestion

---

[7] Note that the *no suggestion* condition was excluded from this analysis since it did not produce emoji suggestions.

mechanisms, and provided more positive feedback on semantic suggestions than the other conditions. Five participants mentioned that semantic suggestions were convenient and timesaving. The convenience might come from the relevance of the semantic suggestions. P13 pointed out, "*The first one [semantic] is better than the second one [lexical], showing more emotion-related emojis. The second one is related to the word itself and it makes no sense to use the emoji in the conversation.*" Although P19 did not use many emojis during the study, she stated that "*their [emojis'] appearance in suggestion bars makes me feel good.*" This feedback supports our finding that people chose more emojis from semantic suggestion than lexical suggestion.

## FIELD DEPLOYMENT

We also conducted a 15-day field deployment to explore the longitudinal effects of the different emoji suggestion systems. This study focused on the usability of the emoji suggestion systems and on their effects on emoji usage during everyday conversations.

### Participants

Eighteen participants (8 females, 10 males) between 18 and 43 years old ($M$=24.0, $SD$=6.4) were recruited via emails, flyers, and word-of-mouth. Inclusion criteria required that participants were able to use English as their primary language and owned a smartphone with Android 6.0 that they used on a daily basis. Those who were in the laboratory study were not allowed to participate in the field deployment due to prior exposure. The 15-day study contained three five-day periods. Participants were compensated $20 USD in the first two periods and $40 for the third, adding to $80 total.

### Procedure

The study was conducted as a partial within-subjects design with the suggestion mechanism as the independent variable. All of the participants used the *no-suggestion* keyboard in the first five-day period as a baseline (however, they could still input emoji from the emoji selection panel). During the second period, half of the participants used the *lexical* suggestion keyboard while the other half used the *semantic* suggestion keyboard. Everyone returned to the *no-suggestion* keyboard during the last period to determine whether they returned to their baseline behavior. In psychology terms, the study compared an ABA condition sequence to an ACA condition sequence.

When participants were enrolled, they were asked to fill out the same questionnaire about online chatting and emoji usage as in the laboratory study. Participants were told that they would be using an emoji suggestion system during the field study, but that they were free to use or ignore the suggestions as they pleased. Participants were instructed to use the keyboard whenever they were typing in English and to keep their phone network connected so they could retrieve emoji suggestion results. The same usage information was logged as before (*Total Characters*, *Total Emojis*, and *Selected Emojis*). After participants signed the consent form, the keyboard was installed on their phone. The keyboard was configured to participants' personal preferences, including its aesthetic theme and vibration behavior.

Participants met with a researcher after each five-day period to have their keyboards reconfigured to another condition and fill out a short questionnaire about the experience (Table 7). After the second period, when emoji suggestions were provided, participants also completed the same usability survey as in the first study (see Table 5).

| Survey After Period 1 |
| --- |
| 1. Do you find yourself using emojis more or less often than before the study? Why? |

| Survey After Period 2 |
| --- |
| 1. How do you like or dislike the suggestion keyboard? Do you find it affecting you (in negative or positive ways) in online communication? |
| 2. Do you find yourself using emojis more often than before the study? Why? |
| 3. Do you have any comments about the keyboard emoji suggestions? |

| Survey After Period 3 |
| --- |
| 1. What do you think of the current keyboard for this period? |
| 2. Do you find yourself using emojis more or less often than before the study? Why? |
| 3. After the whole period, do you have any comments about the keyboard emoji suggestions? |

**Table 7. The survey questions after each period. The emoji suggestions were offered only during period 2, which is why the questions are different for that period.**

## RESULTS OF FIELD DEPLOYMENT

We collected 54 data logs (18 participants × 3 periods), 18 survey results about the usability of emoji suggestions, and 54 open responses analyzed using inductive analysis [16]. As before, *Suggestion* was the independent variable of three levels: *no-suggestion*, *lexical*, and *semantic*.

### Participant Phone Use

Among the 18 participants, 14 stated that they *always* communicate with their phone, three *sometimes*, and one *seldom*. Four participants stated that they *always* use emojis in online conversations, 11 *sometimes*, and three *seldom*. As for the participants' typical emoji entry method, 10 participants manually selected emojis from a list, one participant used lexical suggestions from the keyboard, and seven used both methods.

### Count Measures

The descriptive statistics for *Total Characters, Total Emojis* and *Selected Emojis* per day are shown in  Unsurprisingly, participants used more emojis with lexical and semantic suggestions than with no suggestions. On average, participants who used lexical suggestions in the second period increased their emoji usage by 31.5% over their baseline, while participants who used semantic suggestions increased their usage by 125.1%. We note that the average usage of daily emoji seems low (fewer than 5 emojis per

day). After looking into the data, we found that some participants used over 10 emojis per day, while the other participants used less than one emoji per day.

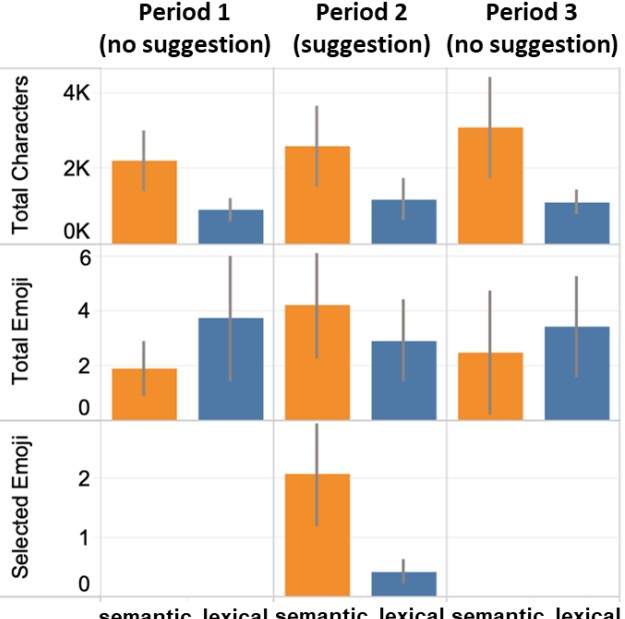

**Figure 5. Averages for *Total Characters, Total Emoji*, and *Selected Emoji* per day from the field deployment dataset. Within each period, the left bar indicates the semantic keyboard group, while the right bar indicates the lexical keyboard group. Error bars represent ±1 standard deviation.**

A Wilcoxon signed-rank test was performed on *Total Characters* and *Total Emojis* between the first and second periods for each group separately. *Total Characters* was not significantly different between the two periods for either *Suggestion* condition. *Total Emojis* was significantly different between the two periods for semantic suggestions ($p<.05$), but not for lexical suggestions. Despite the fact that emoji usage increased in both conditions, only semantic suggestions encouraged participants to input more emojis.

A Mann-Whitney *U* test was performed on *Total Characters, Total Emojis,* and *Selected Emojis* by *Suggestion* for the second period in which the suggestion keyboards were used. The test revealed no significant differences between semantic and lexical suggestions for *Total Characters* and *Total Emojis*; however, semantic suggestions resulted in significantly more *Selected Emojis* than lexical suggestions ($Z$=-2.43, $p<.05$), indicating that those who used semantic suggestions entered a larger proportion of emojis from the suggestion list than from manually picking. This result aligned with findings from our online study.

Furthermore, we analyzed the difference in *Total Emojis* between the different periods by *Suggestion* using Mann-Whitney *U* tests. Results showed that emoji usage increased significantly more with semantic suggestions than with lexical suggestions from the first to second period ($p<.001$). The change between the first and third periods was not

significantly different, indicating that the change in emoji usage was due to the emoji suggestion and not just time.

**Questionnaire Results**
The Likert scale responses from the usability survey during the second period were analyzed using mixed model ordinal logistic regression. No statistically significant differences were found between the semantic and lexical suggestions for any of the questions.

**Discussion of the Field Deployment**
The quantitative analysis results are similar to the in-lab study: the total emoji inputs were similar between different suggestion levels in period 2, and users chose more semantic suggestions than lexical suggestions. Again, based on the survey results, suggestion mechanisms did not influence the online conversation experience significantly.

Participants favored the convenience, enjoyment, and high sentiment relevance of the semantic suggestions. P15 mentioned when her choice of emojis was influenced by the suggestion results: "*I feel that there have been a few instances in which I would use a particular emoji when using a keyboard that was not enabled with emoji suggestion, and when this keyboard suggested a different emoji, I felt that it suited my preferences better.*" P5 even mentioned that he would "*start phrasing the sentences differently to kind of trick the keyboard into predicting the specific emoji I want without having to go to the menu and select it manually.*"

In the group that used lexical suggestions, participants expressed neutral opinions of the suggestion system. Two participants liked the relevance of the suggestions, which entailed providing options after typing a related word. For example, P3 was pleased "*when [lexical suggestion] provided suggestions based on the context of a word, such as smiley faces when typing 'happy'.*" Two participants also enjoyed the various options that lexical suggestions provided. P1 wrote, "*I don't use emojis a lot, but when I do, they're usually in an ironic sort of way. The emoji suggestion keyboard allowed me to do this at times that I didn't think there was a relevant emoji.*" This instance was noteworthy because P1 mentioned ironic emoji usage when there were no relevant emojis he could use in the suggested results.

Comparing the responses after the second and third periods revealed suggestions for ways that the two suggestion mechanisms could be improved. For semantic suggestions, six participants suggested increasing the variety of emoji options. For lexical suggestions, five participants wanted more relevant suggestions. P20 offered a detailed example: "*Sometimes the predicted emoji missed the meaning of what I was typing. For example, when responding to a friend who was apologizing to me, I typed, 'No worries.' I say this in a positive way, however, the emojis suggested were sad or anxious expressions, probably based on the last word typed, which was 'worries'. Therefore, the suggestion missed the intended meaning of the phrase, so maybe it would be impactful to work on the algorithm to detect multiple*

*words/phrases to better understand the meaning within a message*." The above observation is the very reason for why semantic suggestion systems have been proposed in the past [2,9], and also provides supporting evidence of why people picked more semantic emojis in our online study.

## DISCUSSION

Our goal was to examine the impact of emoji suggestion on online conversations. In particular, we sought to answer two primary questions: (1) How do emoji suggestion systems affect the chat experience? (2) Do lexical and semantic suggestion systems affect daily emoji usage differently? We first conducted an online study to evaluate the performance of the two systems, finding that semantic emoji suggestions were perceived as more relevant than lexical emoji suggestions. We then conducted two experiments, finding that emoji usage had a stronger effect on senders than on receivers, but the suggestion system in use did not affect the overall chat experience. A possible explanation is that the suggestion levels only affect the ease of inputting an emoji. Although participants picked more from the semantic suggestions, they could still manually pick their desired emojis if those emojis were not suggested, leading to similar numbers of total emojis inputted with the different suggestion systems.

However, both our in-lab study and our field deployment revealed that the suggestion systems influenced how users reflected on their own experiences. Participants were clearly most excited about semantic suggestions. Even without knowing the details of the different suggestion systems, the participants were pleasantly surprised that the predicted emojis were related to the sentiment of their messages. During the field deployment, participants used more emojis in their daily conversations from semantic suggestions than from lexical suggestions. This finding shows that the semantic suggestions provided more relevant emojis than did the lexical suggestions.

### Design Guidelines for Emoji Suggestion Systems

Based on feedback from the user studies, we created several design guidelines for future emoji suggestion systems:

• *Suggestion Diversity*. Emoji suggestion systems should suggest various types of emojis, ranging from emojis that portray objects to emojis that portray emotions. Although semantic suggestions were preferred in our study, many participants wanted the system to provide more suggestions than just face emojis. Some participants also appreciated that the lexical suggestion system would sometimes suggest rare emojis.

Suggestions from multiple systems could be combined to provide more diverse emojis. Lexical suggestion could provide emojis as the user is typing a sentence, and once the user has finished the sentence, semantic suggestion could provide emojis that reflect the message's overall meaning. Combining the two suggestion schemes could be useful because not all messages contain strong semantic

information, and people also use emojis to provide additional information for certain words [6], such as changing the tone.

• *Personalization*. Beyond providing the most common emoji suggestions, emoji suggestion systems should be aware of the user's personal favorites and usage behaviors. Usage behaviors could be based on categories (*e.g.*, faces, hearts) or the emotions that the user prefers to express. In addition, it would be useful if the suggestion keyboard could recognize the recipient or the usage scenario. For example, a user might want heart emojis when chatting with a family member on a messaging app, but object emojis when composing an email.

• *Avoiding Intrusion.* Emoji suggestion keyboards should only predict emojis when necessary. Some participants only wanted suggestions at the end of messages, as they found the always-on style of semantic suggestions to be distracting.

### Limitations

One limitation of our study is that the suggestion frequency of the two emoji systems was not the same. The semantic suggestion system updated with each new word typed, while the lexical suggestion system updated only after each pre-defined keyword. Thus, participants were exposed to more suggestions in the semantic condition than in the lexical condition. We used our online study to measure the relevance of emoji suggestions independent of frequency. Collecting a similar measure could have been done in our other studies by counting the number of selected emojis and dividing by the total number of emoji suggestions; however, such a metric would neglect many other factors that affect selection rate (*e.g.*, time duration, ordering of emojis).

Another limitation is in our keyboard implementation, namely that the existing semantic-level suggestion model we used contains only 64 possible emojis, thus limiting the diversity of possible suggestions. The *DeepMoji* model could be extended to more emojis, but we chose to stay with the original set to align with the findings from Felbo *et al.*'s prior work [9], since there is no available conversation datasets with emojis for fine tuning the model.

## CONCLUSION

In this work, we compared two emoji suggestion systems: lexical and semantic. Specifically, we explored whether the suggestion type affected the online chat experience and how people perceive the two suggestion types. Our online crowdsourced study revealed that people perceived semantic suggestions as most relevant. Our laboratory study showed that semantic emoji suggestions were used about 1.5 times more than lexical emoji suggestions. Our longitudinal field deployment showed that semantic suggestions led to an increase in emoji usage and were preferred because of their relevance to emotions. As other research in this area has found [6,11,13], we can conclude that emojis themselves, rather than the type of suggestion system, affects the chat experience most profoundly.

Based on our study results, we offered design guidelines for emoji suggestion systems. We believe that by incorporating semantic information in emoji suggestion, researchers can provide better experiences in text-based computer-mediated communications.

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
