# OpenReview forum: "A Comparative Study of Lexical and Semantic Emoji Suggestion Systems"
_graphicsinterface.org/Graphics_Interface/2020/Conference — Submitted to GI 2020_

### Official Review · AnonReviewer2 · 2020-01-07
**On the Fence**

**Confidence:** 4
**Rating:** 6

**Review:**

This paper presents several studies of emoji suggestion systems for  online conversations. The paper includes a preliminary crowdsourcing study, an in-lab study, and a 15-day longitudinal field deployment. In particular, the studies compare a lexical suggestion system (which suggest emoji that match words in the text that has been typed) and a semantic suggestion system (which suggests popular emoji that match the sentiment of the message). The paper also presents design guidelines for emoji suggestion systems.

I'm somewhat torn on this paper. On the one hand, the paper is well written and easy to read; the studies are described in a good amount of detail, with justifications for the different decisions that were made; and the analysis is clear and well presented. On the other hand, I'm not sure that the studies achieve their intended goal of revealing deeper insights into emoji suggestion mechanisms, which weakens the contribution of the work.

Expanding on the latter point above, the main stated contribution of the work is in comparing the lexical and semantic suggestion approaches. However, because the semantic suggestion system suggested popular emoji (while the lexical did not), it's not clear how much of the preference for the semantic suggestion system has to do with the semantic approach itself, versus just suggesting popular emoji (which may be popular because they are relevant in many different contexts). This weakens the main finding of the paper.

A second criticism I have of the paper is that it is unclear how some of the design guidelines follow from the study results. The results seem to favor the semantic approach, but the first design guideline recommends suggestion diversity. The second guideline, personalization based on user and usage context, seems entirely unrelated to the studies that were conducted and their findings.

Finally, the finding that the chat experience was not measurably affected by the suggestion systems is a negative result. This isn't necessarily bad, but it's hard to say what we can take away from it to inform further research or practice.

As a result of the above, I'm on the fence on this paper and have given it a neutral score. If the paper is accepted, I think it should acknowledge that the popularity of the emoji suggested in the semantic condition (as compared to the lexical condition) may have played a role in the results, and the design guidelines should be modified to align more closely with the study findings.

Smaller points:

Page 9 - "We then conducted two experiments, finding that emoji usage had a strong effect on senders than receivers, …" I went back and looked for this finding in the earlier sections, but I couldn't find it. What is this referring to?

Page 5 - "most commonly selected tweet per emoji" - I think this should be "most commonly selected emoji per tweet"

Page 5 - Q4 in Table 4 is cut off.

---

### Official Review · AnonReviewer1 · 2020-01-09
**the paper explores a research question with three types of studies and presents the details of the findings. At the same, I’m concerned about the significance of the research question, some aspects of the experimental design, and the foundations on which the design guidelines are based on. I'm on the fence for the paper toward rejection.**

**Confidence:** 4
**Rating:** 5

**Review:**

The paper compares two emoji suggestion systems (i.e., lexical and semantic) in three ways: a small-scale crowd-sourcing study, a lab study, and a field deployment study. Results show that semantic emoji suggestions were perceived as more relevant than lexical emoji suggestions. However, the suggestion type did not affect chat experience significantly.

The strengths of the paper are as follows: the paper is well-written and easy to follow. The paper contains three types of evaluations: a small-scale crowd-sourcing study, a lab study, and a field deployment study. Details of each study are presented sufficiently. The results of the three studies are complementary to each other.

The weaknesses of the paper are as follows: First, the motivation of the work is the effect of the two emoji suggestion mechanisms on chat experience is unknown. One key finding is that the two emoji suggestion mechanisms did not affect chat experience significantly. The paper only provides a brief conjecture in DISCUSSION that the two types of emoji suggestion mechanisms might only affect the ease of inputting an emoji. Such an explanation (conjecture) is unconvincing. The study should have designed to gain more qualitative data to understand what factors affect their chat experience.

Moreover, the studies found that users entered roughly the same amount of emoji with and without any suggestion mechanisms. This finding challenges the significance of the research question. Why is it an important research question to study in the first place (other than no one has studied this question)?

Second, the frequency of lexical and semantic suggestions is different. One is at word-level and the other is at sentence-level. Even though the authors acknowledge this limitation, such design did introduce a confounding variable into the experiment. Perhaps the word-level suggestion is so frequent that participants felt overwhelmed and tended to ignore them.

Third, the connection between the design guidelines and user studies is ambiguous. For example, it is unclear what the second design guideline is based on even though it seems to be reasonable.

In sum, the paper explores a research question with three types of studies and presents the details of the findings. At the same, I’m concerned about the significance of the research question, some aspects of the experimental design, and the foundations on which the design guidelines are based on. I'm on the fence for the paper toward rejection.

---

### Official Review · AnonReviewer3 · 2020-01-09
**Well written but analysis missing**

**Confidence:** 3
**Rating:** 5

**Review:**

This paper is well-written, an easy read that clearly describes three experiments. I also think that the quantitative data from the experiments was well-done. Finally, the idea of creating a system to simplify emoji use would definitely be an aid -- as anyone who has tried to search through the many emojis available on mobile keyboards.

I wish I knew more about past work in emojis prior to writing this review. While, overall, I am aware of work that explores the benefits of emojis in CMC, I am not fully informed about any studies that look, specifically, at lexical emojis and their overall purpose.

I want to start this review grounded in the paper. At one point, one participant notes that they primary use lexical emojis in an ironic sense, as a humor mechanism, and I believe that there is some truth to that. If one examines past work in emoji use (both cited and via a quick google scholar search), most of the work of which I am aware or that I can find highlights -- as does this paper -- that emojis are primarily useful for their affective nature, the emotional component that they convey during communication. I actually think that this result is also mirrored in the data collected in this paper, particularly some of the qualitative data, as highlighted by the above quote. Lexical emojis like the 'school' emoji serve a limited purpose, particularly as suggestions, when I decide that I'm going to pick up my kids after school. The replacement of school with an emoji, to me, makes little sense in this context, as, as noted in the paper, the word has already been typed. This is at odds with semantic emojis, where the affect hasn't been explicitly indicated, and the emoji can serve as a useful tool to add this affective channel to communication.

What is most interesting to me in this work is the 'crossed wires' effect that I see, where it almost seems like lexical emojis foster increased use of semantic emojis. Table 6 and Figure 5 highlight this for me. In Table 6, we see emoji use jump for both semantic and lexical, but it doesn't necessarily seem that people are selecting from the lexical emojis, and, in Figure 5, we also see this confound where people select the semantic emoji, but the selected emoji percentage is really low (might be a "why would I replace the word I just typed with an emoji thing"). I can't help but wonder what would happen if the system simply suggested random semantic emojis (e.g. one happy and one sad) for every message, just to see what happens. Basically, I'm wondering if showing emoji suggestions makes a writer remember to consider adding emojis, and, for that, they simply add the emojis in.

All this being said, what I feel is missing in the paper is this completeness in results. We see that both lexical and semantic emojis increase emoji use, but many other measures are unrevealing, and there is this question, even with users who a priori used lexical emojis, of whether those lexical suggestions helped. I would love to see more information, particularly from the lexical emoji users, that seeked to probe why their use of emojis increased even when their selection from suggested emojis did not. Were they typing semantic emojis?

Overall, my take-away at this point from the paper is this mixed message, but not an analysis of this mixed message of lexical suggestion but not selection confounded with increased emoji use. This observation is interesting, but I'm not sure, without some drilling down, that it is enough to carry the paper in its present form.

If I were to make a suggestion to the authors, my suggestion, depending on whether this paper is accepted or not, would be, if unsuccessful, to spend some time on the qualitative data. I would really like to have this data set more fully developed in the paper, beyond the relatively surface treatment of quotes in the paper. I would also suggest that the authors consider a study with random semantic emojis (as opposed to the ones produced algorithmically) to see if it is, perhaps, the prompting that emojis are possible that changes emoji use.

---

### Meta-Review · Area_Chair1 · 2020-01-10

**Recommendation:** Reject
**Confidence:** 4

**Metareview:**

All of the reviewers appreciated the clear presentation of this work, finding it to be well-written with clear justifications for the design of the presented studies. The reviewers also appreciated that the paper includes a number of different studies to probe its research questions.

Though the presentation was appreciated, all of the reviewers also raised concerns about the paper. R2 and R1 expressed concern that there may be confounds in the comparison of semantic and lexical systems (due to the popularity of the emoji in the semantic system, and the different mechanisms through which suggestions are made). R2 expressed concerns about the contribution of the work being thin, in part because it is unclear what is driving the preference for the semantic suggestion system. Finally, R1 and R2 both pointed out that the design guidelines are not clearly linked to the results of the studies.

In terms of how this paper could be improved, R1 and R3 suggested that the paper would benefit from a more in-depth analysis of qualitative data, to gain deeper insights into the different suggestion methods. I agree with this suggestion, and I think that if it was done well, the insights from such an analysis would strengthen the qualitative results, and also enable the paper to make design recommendations that are more grounded in findings (that is, it could address many of the criticisms raised above).

As it stands, my recommendation is Reject, because I think that the paper as it stands does not make enough of a contribution, and the addition of a more in-depth qualitative analysis would represent a major change that would need a new review cycle to evaluate.

---

### Decision · Program_Chairs · 2020-01-11

Reject